# A Single-Button Mobility Platform for Cause–Effect Learning in Children with Cerebral Palsy: A Pilot Study

**DOI:** 10.3390/children12081077

**Published:** 2025-08-16

**Authors:** Alberto J. Molina-Cantero, Félix Biscarri-Triviño, Alejandro Gallardo-Soto, Juan M. Jaramillo-Pareja, Silvia Molina-Criado, Azahara Díaz-Rodríguez, Luisa Sierra-Martín

**Affiliations:** 1Departament of Electronic Technology, Universidad de Sevilla, Escuela Politécnica Superior, 41011 Sevilla, Spain; fbiscarri@us.es (F.B.-T.);; 2Department of Continuum Mechanics and Theory of Structures, Universidad de Sevilla, Escuela Politécnica Superior, 41011 Sevilla, Spain; 3Department of Design Engineering, Universidad de Sevilla, Escuela Politécnica Superior, 41011 Sevilla, Spain; 4Department of Physiotherapy, ASPACE, Dos Hermanas, 41704 Sevilla, Spain

**Keywords:** mobility platform, cause–effect learning, severe disabilities, reaction times, keystroke accuracy

## Abstract

**Highlights:**

**What are the main findings?**
A wheelchair-mounted, semi-autonomous mobility platform operated via a simple switch is feasible and usable for children with severe motor disabilities (GMFCS IV–V).Preliminary data indicate a positive trend in reaction times and keystroke accuracy with the use of the platform; however, the small sample size limits the statistical robustness of these findings.

**What is the implication of the main finding?**
The platform could help enhance engagement in cause–effect learning tasks for children with severe motor impairments.The study provides essential baseline data to design larger, statistically powered trials assessing the platform’s impact on causal.

**Abstract:**

**Background:** Mobility plays a fundamental role in causal reasoning (causal inference or cause–effect learning), which is essential for brain development at early ages. Children naturally develop causal reasoning through interaction with their environment. Therefore, children with severe motor disabilities (GMFCS levels IV–V), who face limited opportunities for interaction, often show delays in causal reasoning. **Objective:** This study investigates how a wheelchair-mounted, semi-autonomous mobility platform operated via a simple switch may enhance causal learning in children with severe disabilities, compared with traditional therapies. However, due to the scarcity of participants who meet the inclusion criteria and the need for long-term evaluation, recruitment poses a significant challenge. This study aims to provide an initial assessment of the platform and collect preliminary data to estimate the required sample size and number of sessions for future studies. **Methods:** We conducted a pilot randomized controlled trial (RCT) to assess platform usability and its effect on reaction time and keystroke accuracy. Four children, aged 8.5 ± 2.38, participated in seven 30 min sessions. They were randomly assigned in equal numbers, with two participants in the intervention group (using the platform) and two in the control group (receiving standard therapy). Usability was evaluated through a questionnaire completed by two therapists. Key outcome measures included the System Usability Scale (SUS), reaction time (*RT*), and keystroke accuracy (*NIS*). **Results:** Despite the small sample size and recruitment challenges, the data allowed for preliminary estimates of the sample size and number of sessions required for future studies. Therapists reported positive usability scores. Children using the platform showed promising trends in *RT* and *NIS*, suggesting improved engagement with cause–effect tasks. **Conclusions:** The findings support the feasibility and usability of the mobility platform by therapists, although some improvements should be implemented in the future. No conclusive evidence was found regarding the platform’s effectiveness on causal learning, despite a positive trend over time. This pilot study also provides valuable insights for designing larger, statistically powered trials, particularly focused on *NIS*.

## 1. Introduction

Locomotion is essential for psychological growth. It emerges early in life, coinciding with rapid neuro-cognitive development, indicating a strong link between these phenomena [1]. Hence, individuals with disorders affecting mobility, posture, or muscle tone, such as cerebral palsy [2], may show early cognitive delays.

A fundamental aspect of this developmental process is cause–effect learning [3], or simply causal learning. Many areas of the brain are involved in this process. For example, the prefrontal cortex (PFC) helps predict the outcomes of actions and evaluate their consequences [4]. Specifically, the dorsolateral prefrontal cortex (DLPFC) is particularly important for integrating information and making decisions based on causal inference [5]. Other regions, such as the hippocampus, play a key role in forming and retrieving memories, which is essential for associating causes with their effects over time [6]. The basal ganglia, including the striatum, are involved in habit formation and procedural learning, which help establish associations between actions and rewards, reinforcing behaviors that lead to desired outcomes [7]. Furthermore, the parietal cortex is involved in spatial awareness and attention, which are important for tracking the relationship between actions and their effects in the environment [8]. Finally, the cerebellum also contributes by refining actions based on feedback from their effects [9]. All these brain regions work together to enable individuals to learn from experience, anticipate the consequences of their actions, and adapt their behavior accordingly.

Children naturally acquire cause–effect understanding by interacting with objects or seeking comfort from caregivers. Perception, emotions, memory, and attention all contribute to this process. However, many children with cerebral palsy rely heavily on family members, caregivers, or therapists to initiate any action, which severely limits their interaction with the environment. This restriction reduces the sensory experiences necessary for normal neural development. To support toddlers in achieving typical causal reasoning development, therapies often involve tasks where the child must perform an action (a cause) that, when executed correctly, results in a rewarding outcome (an effect). For instance, an adapted computer game that periodically stops and requires the toddler to press a button to continue could serve this therapeutic purpose.

Alternatively, increasing mobility and independence during early childhood can lead to more typical brain development. Studies have shown that using a powered wheelchair before the age of 6 positively affects social skills [10], enhancing both communication abilities and interaction with the environment. Similarly, modified electric toy cars for children with cerebral palsy have been shown to improve motor skills, mobility, and social engagement [11]. Literature reviews indicate that such interventions in young children (under age 6) foster oral communication and group participation [11], with even greater cognitive effects reported when implemented before age 3 [12].

In a study by Agrawal [13], children with cerebral palsy aged 1 to 43 months were evaluated to determine their ability to operate a powered wheelchair, maintain interest, and demonstrate progress. Significant improvements in cognitive and motor skills were observed after 30 sessions (20 min each, twice a week). While early intervention is key, similar studies with older children have also shown changes in brain structure. Kenyon et al. [14] trained children aged 3 to 12 to use powered wheelchairs while monitoring brain activity, finding changes in theta and alpha bands in the frontal–parietal regions over time.

Most of these studies involved children operating a wheelchair with a joystick in a controlled environment, which assumes specific manual and visual capabilities. Only a few studies, such as that by Jones [12], included children with severe disabilities (GMFCS levels IV–V) [15], who used proximity sensors or head movements to control the wheelchair.

Our proposal extends assistive mobility solutions to children with severe disabilities (GMFCS levels IV–V) who are unable to operate a joystick. We developed a platform that can be mounted beneath a wheelchair, enabling semi-autonomous indoor navigation with safety features. A software application running on Android devices allows for the collection of experimental data. Common assistive switches or buttons, placed near body parts with voluntary movement (e.g., arms or head), are used to control the platform. The navigation process requires frequent button presses, thereby promoting cause–effect learning. We hypothesize that our proposed system will enhance causal reasoning more effectively than traditional therapies. Its key advantages include offering a more engaging and enjoyable experience, and exposing users to dynamic stimuli directly related to their movements—factors that are essential for developing cause–effect understanding.

However, this study presents a major challenge; recruiting a sufficient number of participants who meet the inclusion criteria for robust statistical analysis is difficult. Specifically, considering the prevalence of cerebral palsy in the general population in Spain ranges from 2 to 2.5 per 1000 births (https://aspace.org/algunos-datos accessed on 10 June 2024), which aligns with rates reported in other high-income countries [16], and that the percentage of cases corresponding to GMFCS levels IV and V ranges between 34% [17] and 28.6% [18], combined with a birth rate of approximately 1.2 per 100 (https://data.worldbank.org/indicator/SP.DYN.TFRT.IN?locations=ES accessed on 10 June 2024), and the fact that the study takes place in a city with a population of approximately 0.7 million (https://sevillapedia.wikanda.es/wiki/Portada accessed on 10 June 2024), the estimated target population size is just 32 children. Even if the entire population were included, the study might still yield non-significant results, depending on the desired statistical significance level, statistical power, and effect size [19]. Typical values of 0.05 for the significance level and 0.8 for power are used in such cases. The effect size, defined by Cohen’s d in hypothesis testing, depends on prior knowledge of the means and standard deviations of the control and intervention groups—data that is currently lacking in the literature.

Based on the considerations above, this study not only aims to assess the proposed framework but also to determine the necessary sample size for a robust statistical analysis. This will help evaluate the effectiveness of the platform compared to traditional cause–effect therapies for children with severe mobility impairments.

Below is the list of research questions.

RQ1What is the usability of the platform? Are there aspects that must be improved?RQ2What are the typical values of reaction times and keypress accuracy (measured as the number of incorrect selections) for both the mobile platform and traditional therapies?RQ3What is the sample size needed to validate whether the use of a mobility platform improves cognitive capabilities (cause–effect learning) in terms of reaction times and keypress accuracy?RQ4How many sessions are required to achieve robust test validation over time?

## 2. Methods

### 2.1. Ethics Information

All participants or their legal tutors received an information sheet outlining the study’s main objectives and the application procedure in detail. Those who agreed to participate in the study were required to sign a consent form before the experiment began. Participants had the option to withdraw from the study at any time and request the removal of their associated data. Data access will always be restricted, and the identities of participants will never be disclosed.

This study has received approval from the Ethics Committee of the Universidad de Sevilla on 22 June 2022, with an internal protocol code of PEIBA 0669-N-22.

### 2.2. Participants

There are two profiles of participants: children and therapists. Therapists should be members of the ASPACE staff and typically engaged in cause–effect therapy with the children. ASPACE is a center that attends to people with cerebral palsy in the metropolitan area of Seville.

Children must meet the following inclusion criteria:Diagnosed with cerebral palsy.Classified as GMFCS level IV or higher, indicating an inability to walk or operate a powered wheelchair.Under 10 years old.They can understand simple instructions (like press a button to make the platform move and wait for the platform to stop to press the button again).Must be affiliated to a center whose facilities guarantee a proper navigation experience.

Two therapists and four children were enrolled in this study. The children will be identified as PC, PV, PT, and PR. They are all associated with ASPACE, in which we could conduct the experiment in a 100-square-meter hall with scarce furniture and architectural elements.

PC is a 10-year-old boy with spastic tetraparesis affecting all four limbs (both upper and lower) and GMFCS IV. He experiences epileptic seizures that were previously controlled, but have become more frequent recently, leading to a neurological review to adjust medication. He has good language comprehension but difficulty articulating words; he is currently working on this aspect through Alternative and Augmentative Communication (AAC) using a communicator.

PV is a 10-year-old girl suffering from spastic tetraparesis with bilateral right-sided involvement, affecting both upper and lower limbs, but with good functionality in the left upper limb and GMFCS IV. She has uncontrolled epileptic seizures and behavioral problems that affect her learning. Currently, she is working on Alternative and Augmentative Communication (AAC) therapy using a communicator.

PR is a 5-year-old boy. The probable genetic origin is not yet diagnosed; tests are still being conducted. To date, it is consistent with cerebral palsy. He has hypertonia of the limbs (greater involvement of the lower limbs) and GMFCS IV. He has severe to moderate intellectual disability.

PT is a 9-year-old boy with spastic tetraparesis affecting all four limbs (both upper and lower) and GMFCS IV. Cognitively, he is very well connected to his environment, with a very good level of language comprehension, although he has difficulties expressing himself and is working on this through a communicator.

As described above, all participants have a high level of disability, which limits their ability to communicate verbally and to manipulate devices such as tablets and computers. They interact with these devices using adapted tools, such as buttons.

### 2.3. Design

Therapists completed the SUS questionnaire (Appendix A), along with supplementary questions (Appendix B), after the seventh session to evaluate the mobility platform. The supplementary questionnaire included two 5-point Likert scale questions assessing the platform’s potential to enhance social skills, as well as two open-ended questions regarding suggested improvements.

The children were randomly assigned to two equal-sized groups: the *control* group, in which participants continued with their regular therapy, and the *intervention* group, in which participants used the platform. Group assignment was performed using a computer-generated random number sequence. Participants were blinded to their group allocation, but blinding the therapists administering the interventions was not feasible. However, data analysis was carried out by a researcher who was blinded to the group assignments. PC and PV were assigned to the intervention group, while PT and PR were assigned to the control group. Some traditional cause–effect therapies involve a video that pauses periodically, requiring the child to press an adapted button to resume playback. Both the control and intervention groups followed a structure that mimicked this form of therapy.

Figure 1 shows the experimental timeline, which consisted of seven weekly 30 min sessions. Prior to the intervention, therapists received training on how to use the platform and the configuration app. They had the opportunity to become familiar with the system during the two weeks preceding the experiment.

During the sessions, children in the intervention group used the platform, while those in the control group continued with their traditional therapy. The researchers ensured that the interaction parameters in the traditional setting were adjusted to match those of the platform as closely as possible, allowing for meaningful comparisons.

In the intervention group, therapists only needed to attach the platform to the wheelchair, position the adapted button, and activate the system. A mobile application enabled remote control of the platform if necessary. Therapists were also allowed to remind participants of the task when they lost focus. In the control group, therapists followed the same procedure as in the intervention group.

### 2.4. Sample Size Estimation

One of the main challenges in any scientific study is specifying the statistical power of a test while maintaining a predefined significance level. To achieve this, an accurate estimation of the required sample size is essential.

Calculating the sample size requires prior knowledge of the means and variances of the dependent variables in both the intervention and control groups. Currently, such data are not available from previous studies. Therefore, this study can serve as a preliminary step to estimate these parameters and determine the sample size necessary to obtain statistically conclusive results.

To estimate the required sample size *n*, we used Equation (Equation 1), where $z_{\alpha }$ is the z-score corresponding to the significance level α (typically set to 0.05), and $\Phi^{-1}$ is the inverse cumulative distribution function (quantile function) used to determine the value corresponding to the desired statistical power 1−β (commonly set to 0.8). The effect size (*ES*), also known as Cohen’s *d*, is defined as the ratio between the difference in means of the two groups and the standard deviation, i.e., ES=μ/σ. Since the numerator of Equation (Equation 1) depends only on α and 1−β, estimating the *ES* requires prior knowledge of the means and variances of the outcome variables.(1)$$n\geq {\left(\frac{z_{\alpha }+\phi^{-1}\left(1-\beta \right)}{ES}\right)}^{2}$$

### 2.5. Materials

A semi-autonomous guidance mobility platform is attached to the children’s wheelchair [20] (Figure 2), providing safe navigation for a predefined period (dwell time of 30 s). The movement commences when the child presses an adapted button. The platform advances and adjusts its direction upon detecting obstacles through sensors. The wheelchair’s movements alter the child’s perception of the environment, offering a visual stimulus.

After the dwell time expires, the platform stops, and the child must press the button again to continue moving. This stop–start procedure replicates the cause–effect training. Reaction time, which measures the time elapsed between the moment the platform stops and when the child presses the button again, was collected for all participants and sessions. Additional variables, as described in Section 2.6, were also collected, transmitted, and stored in a database on a cellphone for further analysis. To achieve this, we utilized an Open Source Frontend SDK called Ionic, capable of deploying hybrid mobile applications based on web development tools such as HTML, CSS, and JavaScript. Ionic apps are designed to run in a browser shell similar to Android’s WebView. Various Capacitor plugins were used to provide the necessary functionality for managing the database and Bluetooth connectivity, essential for data storage and communication purposes. The application screen features minimal information to enhance usability, as shown in Figure 3. It includes several buttons for therapists to enable/disable Bluetooth connections, start/stop recording, and a panic button to halt the platform and abort the experiment. Additionally, users can select their username and the type of experiment from two scroll lists.

Common therapies involve the use of educational resources that require children to press a button repeatedly. For the control group, we opted for the use of fairy tale or cartoon videos on YouTube. The specific videos to watch will be tailored to the user’s preferences. These videos frequently pause and require a child’s action to resume playing. To achieve this and obtain the same dependent variables as in the intervention group, we employed an Arduino Leonardo. The adapted children’s button is connected to the Arduino, which, in turn, is linked to the computer via a USB interface. The computer recognizes the Arduino as a keyboard, emulating the shortcut keys (such as ‘k’ or ‘space’) necessary to control the play/pause of the YouTube video. Figure 3 illustrates the hardware elements used in the control group.

The Arduino controls the play interval and hosts an application that detects the user’s keystrokes. This application transmits the measurements via Bluetooth connection to the same mobile phone application used for the mobility platform. To ensure synchronization between the YouTube video and the Arduino, the therapist initially pauses the video and then presses the ’start recording’ button within the mobile application. This action prompts the Arduino to begin guiding the experiment and providing information.

They watched several videos according to their preferences: ‘Blippi’ (https://www.youtube.com/c/BlippiEspaÃśol accessed on 4 July 2022), ‘*El Reino infantil*’ (https://www.youtube.com/@ElReinoInfantil accessed on 4 July 2022), ‘Paw Patrol’ (https://www.youtube.com/@NickJrEspanol accessed on 4 July 2022) and ‘Santiago of the seas’ (https://www.youtube.com/results?sp=mAEB&search_query=santiago+de+los+mares accessed on 4 July 2022).

Details for building the hardware elements, the firmware, and the mobile application can be downloaded from the following link: https://github.com/aljemoca/Cause-effect-Study-Children-with-CP accessed on 4 July 2022.

### 2.6. Measurement Variables

This section describes the variables and tools used in this study. Below is the list of quantitative dependent variables.

On the side of the therapist, we have:SUS test [21] in its Spanish version to measure the ease of use of the proposed framework.Questions Q2 and Q3 of the questionnaire.

On the side of children:Reaction time (*RT*). As explained previously, in the intervention group, *RT* measures the time elapsed between the platform stopping and the user pressing the button to initiate a new navigation period.In the control group, *RT* measures the time elapsed between the video pausing and the user pressing the button to continue watching it.Number of incorrect selections (*NIS*). It represents the number of additional button presses recorded while the platform is in motion and serves validation purposes. A high value of this variable, consistently maintained over time, suggests that the user is not performing the cause–effect experiment effectively.In controls, *NIS* represents the number of additional button presses while the video is playing.

### 2.7. Statistical Analysis

Characterization of dependent variables: Means, standard deviations, normality, and homoscedasticity were assessed. The Shapiro–Wilk test was applied to evaluate normality for each subject, and the Levene test was used to assess variance homogeneity across sessions.Pearson Regression: Pearson regression analysis was conducted to identify trends in the dependent variables over the sessions. This parametric method evaluates the linear relationships between variables, determining how well changes in one variable predict changes in another.Kruskal–Wallis Test: The Kruskal–Wallis test was employed to analyze statistical differences in the dependent variables across sessions. The significance level for this test was set at 5%, meaning differences were considered statistically significant if the *p*-value was less than 0.05.Sample size estimation, n: The sample size was estimated using Equation (Equation 1), based on the confidence level, statistical power, and effect size (*ES*).Number of sessions (*Ns*). Similar to sample size estimation, the number of sessions (*Ns*) can be estimated from the slope and residuals of the dependent variables across sessions. The number of sessions is derived as the sample size using Equation (Equation 1), where the *ES* is defined by the slope and residuals.

## 3. Results

Regarding research question RQ1, therapists believe that using the platform will have a positive impact on children (Q2: 4 out of 5) and will enhance their social capabilities (Q3: 4 out of 5). They also mentioned that the platform should incorporate more sensors, to increase navigation capabilities in more difficult environments, and be compatible with a larger number of wheelchair models (Q1). The last question, Q4, was left blank.

The results of the System Usability Scale (SUS) yielded the following scores: 55 and 62.5. This indicates that the tool is considered ‘Acceptable’ according to [21].

Figure 4 and Figure 5 show preliminary results for *RT* and *NIS*, respectively, where upper plots represent the control group (research question RQ2). We applied the percentile method to remove outliers, which means that any value out of the range of [r1, r2], is considered an outlier. Here, r1 = q1 − 1.5 ∗ irq, r2 = q3 + 1.5 ∗ irq, q1 and q3 are the first and third quartiles, and irq is the interquartile interval (iqr = q3 − q1).

Although it appears that participants in the intervention group show some learning effect over time, on *RT* and *NIS* over sessions, we actually did not find any significant values. This was confirmed by analyzing the confidence interval of the Pearson regression slope and by applying the Kruskal–Wallis test to all participants, PR (*p* = 0.9497), PT (*p* = 0.7402), PV (*p* = 0.9387), and PC (*p* = 0.9207). The same situation applies for *NIS*, PR (*p* = 0.9980), PT (*p* = 0.9983), PV (*p* = 0.8881), and PC (*p* = 0.6116).

The Levene test for all participants demonstrated that the variance follows similar figures over sessions. Additionally, the Shapiro–Wilk test showed that the distribution of measures did not follow a normal distribution.

We then unified data for every participant over time in the control and intervention groups. Table 1 contains averages and standard deviations of these unified *RT* and *NIS*. One participant showed an abnormally high *RT* average compared to the counterparts in the experiment, and, for this reason, we excluded PV from the following analysis.

The *ES* for *RT* between the two groups (research question RQ3) is given by the difference between the averages divided by the standard deviation (Equation (Equation 2)). A similar formula can be applied to determine the *ES* for *NIS*. The reaction times (*RT*s) for subjects PR, PT, and PC were very similar, and around 6.6 s with standard deviations of 5.51 s on average. This means that the effect size, *ES*, is very low, 0.027. Conversely, higher *ES* values were obtained in *NIS* because of the average difference and lower variances. For the sample size estimation, n, Equation (Equation 1) is applied. Table 2 contai*NS* the results obtained.(2)$$ES=\frac{{\bar{RT}}_{c}-{\bar{RT}}_{i}}{\sqrt{\frac{\sigma_{RT_{c}}^{2}+\sigma_{RT_{i}}^{2}}{2}}}$$

Identically to the sample size estimation, the number of sessions, *Ns*, (research question RQ4), can be estimated from the slope and residuals of dependent variables over sessions. The number of sessions is then the sample size given by Equation (Equation 1) and the *ES* obtained by Equation (Equation 3) when the *ES* is defined by slope and residuals.(3)$$ES=\frac{slope}{Residuals}$$

According to the results, the theoretical number of sessions required to achieve statistically significant outcomes is too high to be practical for implementation. In other words, experiments that analyze the significance of this dependent variables must be focused on doing it in only one session.

## 4. Discussion

Mobility platforms, like the one described in this study, aim to foster independence, motor skills, and cognitive engagement while maintaining reliability, intuitiveness, and enjoyment in their use. The SUS test outcomes indicated that the platform’s usability was acceptable. Therapists suggested increasing the number of sensors so that the platform can be used in more environments, and enhancing compatibility with a wider range of wheelchair models to increase the number of potential users.

Excluding one participant, the average reaction times were very similar across subjects, approximately 6.6 s, with a distribution that did not follow normality. The measurement variance was consistent at around 5.45 s and remained stable across sessions for each individual. Conversely, significant inter-individual differences were found, as shown by Levene’s test (*p* < 0.001).

Less uniformity was observed in the dependent variable *NIS*. Differences emerged between the control and intervention groups, with average values of 2.65 for the control group and 1.1 for the intervention group. Similar to *RT*, *NIS* samples did not follow a normal distribution. Variance remained consistent within subjects across sessions but differed between participants (*p* < 0.001).

Three of the four participants (PC, PV, and PT) shared similar characteristics in terms of age, cognitive and communicative abilities, and manual skills. The first two were assigned to the intervention group, while PT and PR were in the control group. PR, the youngest, presented some degree of intellectual delay. However, their *RT*s were similar—excluding PV, who participated after her hydrotherapy sessions, which often left her fatigued and less responsive when the platform stopped. Conversely, PC remained active throughout the experiment, appeared aware of when the platform stopped and started, and expressed enjoyment from the movement.

Differences in *NIS* were more pronounced. PR tended to perform worse, possibly due to his restlessness or cognitive maturity, with frequent button pressing, regardless of the video’s state. He was consistently active, often requested video changes, and seemed easily bored. In contrast, PT was calm, attentive to the videos, and responded appropriately when they paused.

It is well-established that reaction times in individuals with cerebral palsy are generally longer than in typically developing peers. Several studies have investigated techniques to improve *RT* using virtual reality [22] or acute physical exercise [23]. Typical *RT* values in such studies range between 0.6 s and 1.4 s, much lower than the average *RT* observed in this pilot. Importantly, those studies included participants with GMFCS levels I–III and good manual dexterity. By contrast, our participants (GMFCS IV–V) faced considerable challenges in pressing the adapted button due to limited upper limb control. This may have delayed their responses and contributed to the high number of extra presses. Moreover, the variability in *RT* and *NIS* was substantial; *RT* values ranged from a few hundred milliseconds—consistent with previous studies—to over 10 s.

In experimental psychology, reaction times and error rates are behavioral proxies for internal cognitive processes such as causal reasoning. Faster *RT*s and fewer errors typically reflect stronger understanding of the relationship between action and consequence. Conversely, slower or more error-prone responses may indicate confusion, uncertainty, or incomplete reasoning.

Improvements in *RT* and reduced errors may not only suggest enhanced causal reasoning but also highlight the interplay between fine motor skills and technological accessibility. Research has shown that fine motor abilities are strongly linked to early cognitive development and learning outcomes [24]. In children with motor disabilities, accessible technology can help remove physical barriers, enabling a more accurate expression of causal reasoning and learning potential [25].

The effect size (*ES*) for *RT* was very low, due to both the homogeneity between groups and high variability within them. As a result, a very large sample size (>10,000) would be required to detect statistically significant differences in a hypothesis test. As noted in the introduction, the target population in the city where this study was conducted is estimated at around 32 individuals. Thus, attempting to achieve statistical significance for the dependent variable *RT* in this specific population is unfeasible. A more realistic goal emerges when focusing on the *NIS* variable. With an *ES* of 4.51, a sample size of 10 participants would be sufficient to detect significant differences.

This pilot study involved seven sessions with the platform. In the literature, other studies have implemented a greater number of sessions. For instance, ref. [13] examined how children developed driving skills while learning to steer a powered platform using a joystick. They compared short-term and long-term training groups, finding significant improvements only in the long-term group. Similarly, ref. [26] analyzed the effects of low (100–200 min) and high (600–800 min) usage of powered platforms on developmental changes in children with cerebral palsy, showing significant changes only in the high-use group. The seven 30 min sessions in this pilot align with the low-use category of that study. However, based on our estimations, the number of sessions required to statistically validate trends in *RT* and *NIS* would be too high, making such evaluations impractical.

A final remark concerns the platform’s cost and accessibility. In the Materials section, we have included a GitHub link where all software and hardware specifications can be freely downloaded. Although the platform cost (EUR 600–800) is relatively low, it still represents a major disadvantage compared to traditional methods used to support causal reasoning.

Some commercial products offer similar functionalities and can be easily adapted for use with custom buttons. However, the effectiveness of this specific platform has yet to be demonstrated. To justify its adoption, it must show significantly greater benefits than standard therapy.

## 5. Study Limitations

The primary limitation of this study is its small sample size (N=4), which substantially limits the statistical power of the analysis. This constraint hinders the ability to draw definitive conclusions or detect statistically significant differences between groups. As a result, the findings should be considered preliminary and are not sufficient to support conclusive statements.

While we acknowledge that recruiting participants from a single center (ASPACE) introduces a potential selection bias, the center’s broad geographical reach provides access to a more heterogeneous sample than would be expected from a purely local recruitment strategy. Nevertheless, the results must be interpreted with caution, as the participants may not be representative of the broader population of individuals living with this condition.

It is also possible that the reaction time (*RT*) measurements were influenced by participants’ physical limitations, despite the use of adapted switches customized to each participant’s preferred method of interaction. This potential confound could be mitigated in future studies by employing an AB/BA crossover design, where each participant experiences both the intervention and control conditions, thus reducing inter-subject variability and compensating for individual physical limitations.

## 6. Conclusions and Future Work

Locomotion plays a crucial role on the development of the brain during the childhood. Toddlers and children with severe motor disabilities suffer from some neurological delays compared to normally developing counterparts. Therefore, providing these children the capability of moving autonomously through powered platforms might reduce this gap and have a positive impact on their neurological development. In this study we have built a platform that is intended to be used by children with severe disabilities (GMFCS IV–V) through an adapted button. We tested the the usability of the platform and collected preliminary data to estimate the effect size needed for a future robust statistical study on cause–effect learning, measuring two main variables: reaction time and number of extra selections. The experiment followed the scheme of a randomized controlled trial.

We found that long-term studies are not statistically feasible for either *RT* or *NIS* variables. Moreover, finding significant differences in *RT* between groups would require a sample size that is not feasible to recruit. The only remaining dependent variable is *NIS*, which requires at least 10 participants.

In the future, we plan to improve the mobility platform to make it compatible with more wheelchair models and repeat the experiment with a group of at least 10 participants recruited from different centers, following experimental designs that minimize bias between intervention and control groups regarding children skills, such as, for example, an AB/BA scheme, where all participants will use the platform (A) and the traditional system (B) in random order across two 30 min sessions (plus a prior familiarization session).

## Figures and Tables

**Figure 1 children-12-01077-f001:**
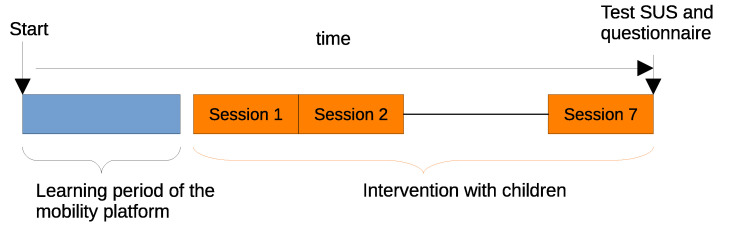
Experimental time schedule. Therapists learn to use the platform before the trials with children. An overall total of seven half-hour-a-week sessions were scheduled in the experiment.

**Figure 2 children-12-01077-f002:**
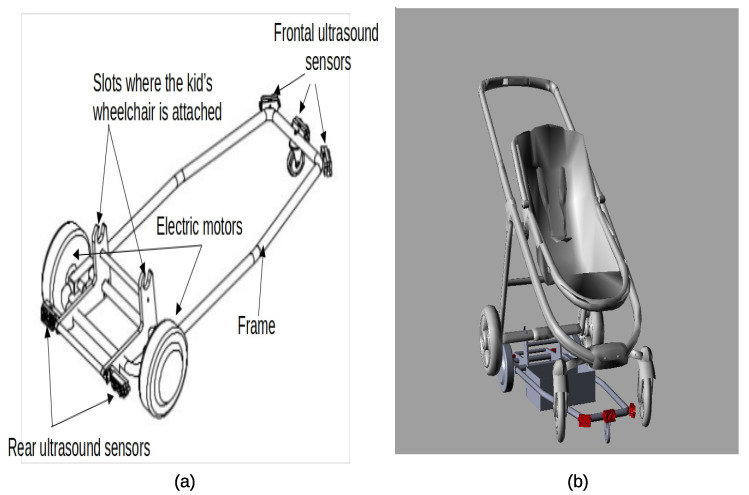
(**a**) A sketch of the mobility platform, which can be attached to most children’s wheelchairs, is depicted. The platform features an ultrasound sensor ring around the frame, enabling guidance and preventing collisions with obstacles. (**b**) Illustration of a wheelchair attached to the mobility platform.

**Figure 3 children-12-01077-f003:**
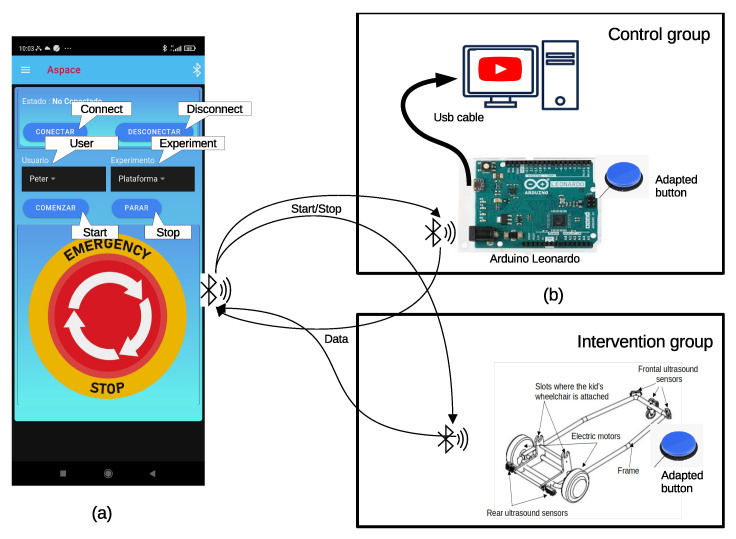
Materials used in experiments. (**a**) Screenshot of the mobile application showing the interface that allows inputting the user’s name, selecting the experiment type, and initiating/stopping data collection. The mobile application is common in both intervention and control groups. (**b**) Elements used in the control and intervention groups. In the former group, an Arduino-based interface was designed to pause Youtube videos every 30 s and convert users’ inputs, through an adapted button, into play commands to resume watching the video. Participants in the intervention group used the platform attached to their wheelchairs. The platform stops moving after 30 s. So children need to press an adapted button to resume the movement again.

**Figure 4 children-12-01077-f004:**
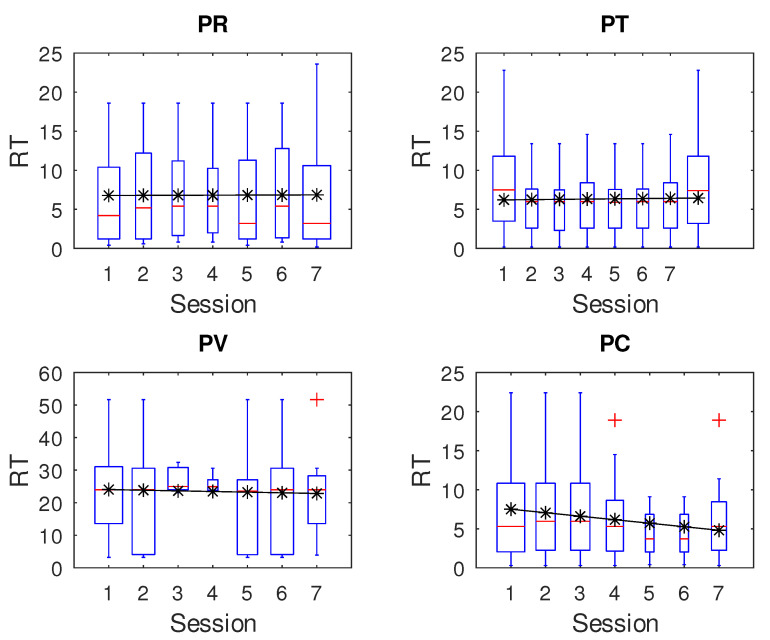
Reaction time over sessions. The upper plots belong to the control group, while the remainders belong to the intervention group.

**Figure 5 children-12-01077-f005:**
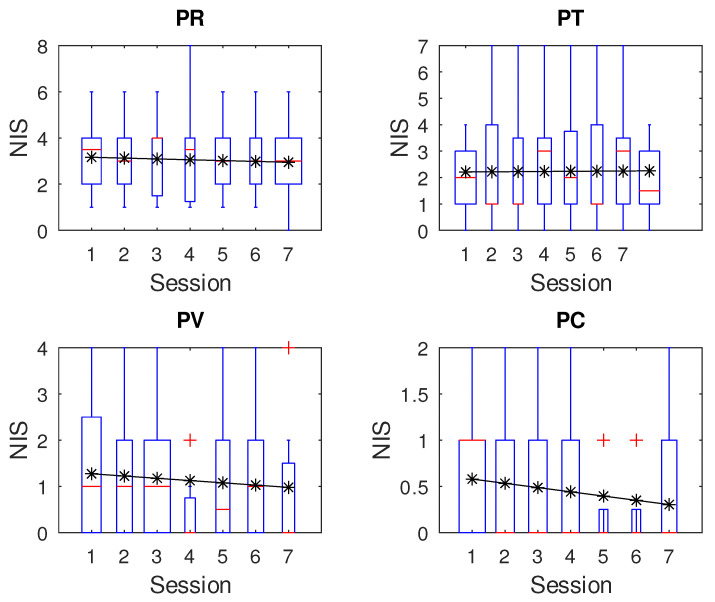
Number of incorrect selections over sessions. The upper plots belong to the control group, while the remainders belong to the intervention group.

**Table 1 children-12-01077-t001:** Averages and standard deviations for *RT* and *NIS*. (*) To estimate *RT* average, PV participant was excluded. The effect size for hypothesis testing and the sample size (n) needed for each group are also shown.

Subject	RT		NIS	
	**Average (s)**	**Std (s)**	**Average**	**Std**
PR	6.7	6.1	3.1	1.6
PT	6.6	4.8	2.2	1.7
PV	23.2	13.6	1.7	0.2
PC	6.5	5.6	0.5	0.2
All	6.6 *	5.51 *	1.73	1.33
Control	6.65	5.45	2.65	1.65
Intervention (*)	6.5	5.6	1.1	0.2
ES	0.027		1.319	
n	>10,000		4.51	

**Table 2 children-12-01077-t002:** Slopes and residuals for *RT* and *NIS*. The effect size for slope validation and the number of sessions, Ns, for each participant are also shown.

	RT				NIS			
**Subject**	**b (s/ses)**	**Residuals**	**ES**	**Ns**	**b (s/ses)**	**Residuals**	**ES**	**Ns**
PR	0.013	6.20	0.003	1.7898 $\times \;10^{6}$	−0.036	2.73	−0.13	45,277
PT	0.033	4.57	0.007	1 $\times \;10^{5}$	0.06	1.86	0.03	7 $\times \;10^{5}$
PV	−0.198	13.85	−0.01	38,519	−0.490	1.47	−0.33	7081
PC	−0.449	5.74	−0.08	1281 $\times \;10^{3}$	−0.46	0.78	−0.59	2227

## Data Availability

Data will be available in the following link: https://idus.us.es/handle/11441/162968 (accessed on 10 June 2024).

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
