# Peer review of "A Single-Button Mobility Platform for Cause–Effect Learning in Children with Cerebral Palsy: A Pilot Study"

_children, 2025, doi:10.3390/children12081077_

Round 1
Reviewer 1 Report
Comments and Suggestions for Authors
Reviewer's comments
Title
- I suggest indicating the type of study
Abstract
- The abstract should explicitly state the number of participants and average age.
- The concept of cause-effect learning is not sufficiently contextualized in the abstract for a clinical reader.
- To indicate more clearly the problem and research gaps.
- The conclusion is somewhat general and could include more specific recommendations for future trials.
Introduction
- The research problem should be established in more detail with the existing gaps in the literature in order to reach the stated objective. A clear hypothesis is not established.
- The authors constantly talk about cause and effect. This should be explained more clearly beforehand.
- There is also constant mention of the improvements and benefits of the interventions. It should be indicated what these interventions are according to the literature.
- It would be useful to clarify whether this type of intervention is intended to replace or complement conventional therapies.
- The introduction should end with the stated objectives and hypotheses. research questions should be integrated with the research problem or hypotheses.
Methods
- It is not specified how the randomization was performed or if there was blinding in the data collection.
- The selection of participants seems to depend exclusively on the ASPACE center, which may introduce a selection bias. It should be clarified.
- Given the limited sample size, I suggest planning the study design as a case study and pilot study.
Results
- The high variability and lack of normality should have led to a more qualitative or mixed approach.
- No results of clinical observation or actual communicative interaction are presented.
- Could high RT values be influenced more by physical limitations (fine motor) than by cognitive learning?
Discussion and Conclusion
- The article should emphasize more clearly that the main finding is technical feasibility and acceptability to therapists, rather than clinically proven efficacy.
- NIS ES is claimed to be relevant, but the clinical relevance of pulsation error as an indirect measure of cause-effect learning is not discussed in depth.
- Expand the discussion section on the relationship between fine motor skills, technological accessibility and learning.
- I suggest providing in subsections the limitations and future lines of the study. The limited sample size should be commented on and be cautious with the conclusions.
- Format of authors' contributions, see journal guidelines.
Author Response
First of all, we would like to thank anonymous reviewers for their valuable comments, and along with the editor for the opportunity of modifying the paper according to their suggestions. Below are the responses to reviewers, the actions taken to fix the issues, and a clear explanation of the manuscript.
Reviewer #1
Comment 1.1 Title I suggest indicating the type of study
Answer: We have modified the title according to reviewers’ suggestions.
Comments 1.2 Abstract
The abstract should explicitly state the number of participants and average age.
The concept of cause-effect learning is not sufficiently contextualized in the abstract for a clinical reader.
To indicate more clearly the problem and research gaps.
The conclusion is somewhat general and could include more specific recommendations for future trials.
Answers: We have added the average age and standard deviation in the abstract, while the number of participants was already included in the previous version.
Additionally, the concept of cause-effect learning has been further contextualized to improve its clarity and relevance for a clinical audience.
Overall, the abstract has been revised to more clearly highlight the problem, existing research gaps, and to provide more specific conclusions and implications for future trials.
Comments 1.3 Introduction
The research problem should be established in more detail with the existing gaps in the literature in order to reach the stated objective. A clear hypothesis is not established.
The authors constantly talk about cause and effect. This should be explained more clearly beforehand.
There is also constant mention of the improvements and benefits of the interventions. It should be indicated what these interventions are according to the literature.
It would be useful to clarify whether this type of intervention is intended to replace or complement conventional therapies.
The introduction should end with the stated objectives and hypotheses, research questions should be integrated with the research problem or hypotheses.
Answers: We have restructured the Introduction section to address your suggestions. Specifically, we have elaborated on the research problem by identifying existing gaps in the literature that justify our study and lead to the stated objectives.
To clarify the concept of cause-effect learning, we have introduced related terms such as causal learning, which may be more familiar to professionals in the field.
We have also explicitly stated the main hypothesis of the study, which proposes an alternative approach to traditional therapies. In addition, we clarified whether the intervention is intended to complement or replace conventional treatments.
Finally, we have revised the end of the Introduction to clearly present the research objectives and hypotheses. The research questions are now integrated with the problem statement and the proposed hypothesis. We have also acknowledged the challenges in achieving a statistically robust sample size.
Comments 1.4 Methods
It is not specified how the randomization was performed or if there was blinding in the data collection.
The selection of participants seems to depend exclusively on the ASPACE center, which may introduce a selection bias. It should be clarified.
Given the limited sample size, I suggest planning the study design as a case study and pilot study.
Answers
The reviewer is right. We have now included a detailed description of the randomization procedure and the blinding process in the manuscript. It is true that selecting participants from the same ASPACE center could introduce a selection bias. However, since this center covers a wide geographical area, this limitation is somewhat mitigated. We have clarified this point in the Study Limitations section.
Regarding the last point, we agree with the suggestion and have revised the title of the study to reflect its nature as a pilot study.
Comments 1.5 Results
The high variability and lack of normality should have led to a more qualitative or mixed approach.
No results of clinical observation or actual communicative interaction are presented.
Could high RT values be influenced more by physical limitations (fine motor) than by cognitive learning?
Answers
This study explores the typical values of certain variables within a very specific population, for which there is little concrete data available. A qualitative or mixed-methods approach is, to some extent, incorporated through the usability test and the questionnaire administered to the therapists, whose results are presented at the beginning of the Results section.
It is also possible that the reaction time (RT) results are influenced by the participants' physical limitations. For this reason, we adapted the switch to the participants' usual mode of use. This issue has been added to the Study Limitations section, where we suggest that AB/BA-type experiments, in which the same subject undergoes both interventions, may better reveal the effect of the platform on reaction times.
Comments 1.6 Discussion and Conclusion
The article should emphasize more clearly that the main finding is technical feasibility and acceptability to therapists, rather than clinically proven efficacy.
NIS ES is claimed to be relevant, but the clinical relevance of pulsation error as an indirect measure of cause-effect learning is not discussed in depth.
Expand the discussion section on the relationship between fine motor skills, technological accessibility and learning.
I suggest providing in subsections the limitations and future lines of the study. The limited sample size should be commented on and be cautious with the conclusions.
Format of authors' contributions, see journal guidelines.
Answers:
Thank you for your comments. We have emphasized more clearly the feasibility and acceptability of the platform from the therapists’ perspective and clarified the clinical relevance of pulsation errors in relation to cause-effect learning. We have also expanded the discussion to outline the relationship between fine motor skills, technological accessibility, and learning.
Additionally, we have added a separate Study Limitations section and addressed the limited sample size, taking a more cautious approach in the conclusions. Finally, we have completed the Authors' Contributions section in accordance with the journal’s guidelines.

Reviewer 2 Report
Comments and Suggestions for Authors
The study sample size is very limited, making generalizability of the results significantly more difficult. The authors may consider including more children. If this is not possible, the study title could be updated to emphasize that it is a pilot study or feasibility study.
The introduction to the study mentions some social effects of locomotion, but only RT and NIS were measured. Evaluations could have been made for outcomes such as "engagement," "motivation," and "emotional response," or this issue could have been addressed in the study's limitations and recommendations for future studies section.
The risk of bias is high in a group with as few as four participants. I recommend that the authors explain the randomization method.
The study reportedly included children who could understand simple instructions. However, it could be clarified how the instructions were given, whether different communication tools were used, or why the children responded by pressing a motor command. Explaining the reason for not using a verbal command or different sensors would facilitate reader understanding.
There are many different studies on locomotion in the literature. Furthermore, the authors' targeting of a sample with severe motor disabilities, such as GMFCS 4-5, is a unique aspect of the study. However, further comparisons with existing systems should be made to emphasize the novelty of the method used.
I recommend that the authors also provide information on the cost and availability of the system they used. The advantages and disadvantages could be presented in a comparative literature review.
Comments on the Quality of English LanguageThere are minor typos in the text. For example, ''tested the usability'', ''platform should incorporate more sensors'', ''seven half-an-hour-a-week sessions'', ''June 22th, 2022'' ) I recommend professional proofreading or reviewing the text for fluency.
Author Response
First of all, we would like to thank anonymous reviewers for their valuable comments, and along with the editor for the opportunity to modify the paper according to their suggestions. Below are the responses to reviewers, the actions taken to fix the issues, and a clear explanation of the manuscript.
Reviewer #2
Comment 2.1 The study sample size is very limited, making generalizability of the results significantly more difficult. The authors may consider including more children. If this is not possible, the study title could be updated to emphasize that it is a pilot study or feasibility study.
Answer: Reviewer is right. We have changed the title to include pilot study and cleared in the text the difficulties in meeting the inclusion criteria even in a wide metropolitan area. For this reason, we include an estimation of the sample size for further studies to demonstrate the benefits of using a platform in causal learning.
Comment 2.2 The introduction to the study mentions some social effects of locomotion, but only RT and NIS were measured. Evaluations could have been made for outcomes such as "engagement," "motivation," and "emotional response," or this issue could have been addressed in the study's limitations and recommendations for future studies section.
Answer: Therapists also filled in a questionnaire where social skills were assessed, although very slightly. The core of the study is mainly focused on RT and NIS.
Comment 2.3 The risk of bias is high in a group with as few as four participants. I recommend that the authors explain the randomization method.
Answer: We have included a brief explanation of the randomization method.
Comment 2.4 The study reportedly included children who could understand simple instructions. However, it could be clarified how the instructions were given, whether different communication tools were used, or why the children responded by pressing a motor command. Explaining the reason for not using a verbal command or different sensors would facilitate reader understanding.
Answer: Children with GMFCS IV and V experience difficulties in controlling their limbs to manipulate mouse pointers, tablets, or similar elements, so the common method of accessing a computer is through adapted buttons or any other technology that converts voluntary actions into binary signals. We have included a paragraph summarizing that oral production is almost inexistent for participants, although they can understand other people. The simplest, cheapest and most usual interacting element is the press button, along with specific applications containing communication boards with ideograms (or characters) and scanning.
Comment 2.5 There are many different studies on locomotion in the literature. Furthermore, the authors' targeting of a sample with severe motor disabilities, such as GMFCS 4-5, is a unique aspect of the study. However, further comparisons with existing systems should be made to emphasize the novelty of the method used.
Answer: There are studies on locomotion in the literature, although this study is focused on cognitive rehabilitation through locomotion, because this is the natural mechanisms by which the brain learns causal reasoning. Our pilot tries to compare the use of plataform with traditional therapies, for this reason we have obtained the same variables using a computer-based cause-effect exercise. This comparison did not give us conclusive outcomes, due mainly to the small sample size. We unknow the existance of similar studies that report these variables for GMFCS 4-5.
Comment 2.6 I recommend that the authors also provide information on the cost and availability of the system they used. The advantages and disadvantages could be presented in a comparative literature review.
Answer: We added such information partially in the discusion section.

Reviewer 3 Report
Comments and Suggestions for Authors
Dear Editor and Authors,
The authors have presented a pilot test of a one-button mobility platform developed to support cause-and-effect learning in children with severe motor disabilities (GMFCS IV–V). The study focuses on a unique topic from both technology development and pediatric rehabilitation perspectives. However, in its current form, the study lacks a sufficiently strong scientific basis and contains numerous methodological and reporting shortcomings. These shortcomings are detailed below. For these reasons, I recommend a "major revision."
The title's phrase "A Novel..." is bold; it does not explain how this innovation is new in the literature.
The abstract section suggests that definitive conclusions have been drawn from a study with only four children; this is scientifically inappropriate.
Although the abstract presents the study as having an effect on cognitive-motor engagement, there is no statistical significance supporting this claim.
The research gap and original contribution are not clearly stated.
The randomization method is unclear; its implementation should be clearly stated.
There was significant heterogeneity among the participating children's characteristics (age, epilepsy, cognitive level, etc.); this may have influenced the analyses and was not adequately discussed.
Information is lacking regarding how therapy practices and platform use were standardized by therapists.
Only numerical scores for the SUS test are provided, and therapist comments are not detailed and are not included in the discussion.
The discussion section engages superficially with the literature and does not include opposing views. A critical approach to the findings is lacking.
The conclusion section includes claims such as "the platform supports neurodevelopmental development," but no such variable was measured in the study. Such statements exceed the limits of inference.
The language of the conclusion section is heavily loaded with technical jargon. Terms such as "Wilcoxon test," "bootstrapping," and "AB/BA crossover design," in particular, appeal to readers with engineering or statistics backgrounds but are not accessible to readers from clinical fields such as child health, physiotherapy, and rehabilitation. The conclusion section should be restructured with simpler language; The platform's potential contribution to the field of child health should be clearly and concisely emphasized.
I find the study valuable due to its originality and its ability to address a significant clinical need in society. However, methodological weaknesses, inadequate sample sizes, and interpretation errors prevent the study from being scientifically convincing in its current form. I believe the authors should carefully review the aforementioned shortcomings and significantly revise the study.
Sincerely,
Author Response
First of all, we would like to thank anonymous reviewers for their valuable comments, and along with the editor for the opportunity to modify the paper according to their suggestions. Below are the responses to reviewers, the actions taken to fix the issues, and a clear explanation of the manuscript.
Reviewer #3
The authors have presented a pilot test of a one-button mobility platform developed to support cause-and-effect learning in children with severe motor disabilities (GMFCS IV–V). The study focuses on a unique topic from both technology development and pediatric rehabilitation perspectives. However, in its current form, the study lacks a sufficiently strong scientific basis and contains numerous methodological and reporting shortcomings. These shortcomings are detailed below. For these reasons, I recommend a "major revision."
Comment 3.1 The title's phrase "A Novel..." is bold; it does not explain how this innovation is new in the literature.
Answer: We removed the “novel” from the title to avoid confusion. This device is not exactly new in the literature, but it is the first time that it is used for cause-effect learning and by people with severe disabilities at our knowledge.
Comment 3.2 The abstract section suggests that definitive conclusions have been drawn from a study with only four children; this is scientifically inappropriate.
Answer: We have rewritten the abstract and agree that, with only four participants, it is not possible to obtain conclusive outcomes. We highlighted the challenge of recruiting children who meet the inclusion criteria, which will require extending the research to other cities. For this reason, we shifted the focus of the study toward obtaining preliminary results and estimating the sample size needed for a robust statistical comparison. According to our findings, it appears that for reaction times, conducting a new experiment may not be feasible, as the required sample size is too large to be realistically implemented. Moreover, as recommended by reviewer #1, we have emphazied the usability of the platform under the therapists’ perspective.
Comment 3.3 Although the abstract presents the study as having an effect on cognitive-motor engagement, there is no statistical significance supporting this claim.
Answer: We have changed the abstract to be more precise regarding this aspect.
Comment 3.4 The research gap and original contribution are not clearly stated.
Answer: Following indications of reviewer #1 we have changed abstract and introduction to make it clearer.
Comment 3.5 The randomization method is unclear; its implementation should be clearly stated.
Answer: Following indications of reviewers #1 and #2 we have added the procedure followed for randomization.
Comment 3.6 There was significant heterogeneity among the participating children's characteristics (age, epilepsy, cognitive level, etc.); this may have influenced the analyses and was not adequately discussed.
Answer: You are right. We have included a paragragh discussing the effect of age and cognitive level on outcomes.
Comment 3.7 Information is lacking regarding how therapy practices and platform use were standardized by therapists.
Answer: We have described the procedures followed by therapists during the experiment.
Comment 3.8 Only numerical scores for the SUS test are provided, and therapist comments are not detailed and are not included in the discussion.
Answer: The therapists comments were included in results. There was only a question that left blank. The discussion contains now a sentence about these comments.
Comment 3.9 The discussion section engages superficially with the literature and does not include opposing views. A critical approach to the findings is lacking.
Answer: We have included some disadvantages of our proposal. Nevertheless, the study is novel in its main objective and for the target population and we did not find similar works to compare at the moment.
Comment 3.10 The conclusion section includes claims such as "the platform supports neurodevelopmental development," but no such variable was measured in the study. Such statements exceed the limits of inference.
Answer: We meant that there was an trend over time in the control group, but, we agree that this comment exceed the limits of inference because the outcomes were not statistically conclusive. Therefore, we have removed this statement.
Comment 3.11 The language of the conclusion section is heavily loaded with technical jargon. Terms such as "Wilcoxon test," "bootstrapping," and "AB/BA crossover design," in particular, appeal to readers with engineering or statistics backgrounds but are not accessible to readers from clinical fields such as child health, physiotherapy, and rehabilitation. The conclusion section should be restructured with simpler language; The platform's potential contribution to the field of child health should be clearly and concisely emphasized.
Answer: We have removed and simplified the technical jargon.
I find the study valuable due to its originality and its ability to address a significant clinical need in society. However, methodological weaknesses, inadequate sample sizes, and interpretation errors prevent the study from being scientifically convincing in its current form. I believe the authors should carefully review the aforementioned shortcomings and significantly revise the study.
Thank you for your interest in this study and for the suggestions.

Round 2
Reviewer 1 Report
Comments and Suggestions for Authors
Most of the reviewer's suggestions have been addressed. Thank you
Author Response
Comment: Most of the reviewer's suggestions have been addressed. Thank you
Response: Thank you so much for your thorough review, which has significantly improved the manuscript.
Reviewer 2 Report
Comments and Suggestions for Authors
I would like to emphasize that your revisions are appropriate and thank you for taking my suggestions into account and implementing them verbatim. These changes have significantly improved the scientific integrity and presentation quality of your article.
Author Response
Comments: I would like to emphasize that your revisions are appropriate and thank you for taking my suggestions into account and implementing them verbatim. These changes have significantly improved the scientific integrity and presentation quality of your article.
Response: Thank you for your suggestions, which have undoubtedly made this manuscript clearer and more scientific.
Reviewer 3 Report
Comments and Suggestions for Authors
The authors have satisfactorily addressed all previous comments and revision requests. The revised manuscript shows clear improvements in methodological clarity, data presentation, and the integration of relevant literature in the discussion section. In its current form, the study meets the journal’s scientific standards and is suitable for publication. I recommend acceptance.
Author Response
Comment: The authors have satisfactorily addressed all previous comments and revision requests. The revised manuscript shows clear improvements in methodological clarity, data presentation, and the integration of relevant literature in the discussion section. In its current form, the study meets the journal’s scientific standards and is suitable for publication. I recommend acceptance.
Response: Thank you for considering this manuscript suitable for publication, and thank you for all your suggestions, which have significantly improved this work.